# Navigating groundlessness: An interview study on dealing with ontological shock and existential distress following psychedelic experiences

Eirini K. Argyri[1]*, Jules Evans[2], David Luke[3], Pascal Michael[3], Katrina Michelle[4], Cyrus Rohani-Shukla[5], Shayam Suseelan[3], Ed Prideaux[6], Rosalind McAlpine[7], Ashleigh Murphy-Beiner[8], Oliver C. Robinson[3]

1 University of Exeter - School of Psychology, Exeter, United Kingdom of Great Britain and Northern Ireland, 2 University of London - Centre for the History of the Emotions at Queen Mary, London, United Kingdom, 3 University of Greenwich - School of Human Sciences, London, United Kingdom, 4 New York University (NYU) - Department of Applied Psychology, New York, NY, United States, 5 Imperial College London, London, United Kingdom, 6 Perception Restoration Foundation, San Juan, PR, United States, 7 University of College London - Division of Psychology & Language Sciences, London, United Kingdom, 8 University of London - Royal Holloway, London, United Kingdom

* e.k.argyri@exeter.ac.uk

## Abstract

Psychedelic induced mystical experiences have been largely assumed to drive the therapeutic effects of these substances, which may in part be mediated by changes in metaphysical beliefs. However, there is growing evidence that psychedelic experiences can also trigger long lasting distress. Studies of persisting difficulties suggest a high prevalence of ontological challenges (related to the way people understand reality and existence). We conducted semi-structured interviews with 26 people who reported experiencing existential distress following psychedelic experiences. We explored the phenomenology of participants' difficulties and the ways they navigated them, including what they found helpful and unhelpful in their process. Thematic analysis revealed that participants experienced persistent existential struggle, marked by confusion about their existence and purpose and preoccupation with meaning-making. Along with cognitive difficulties stemming from the ungrounding of their prior frameworks for understanding, participants' ontologically challenging experiences also had significant emotional, social, bodily and other functional impact. Participants managed to alleviate their distress primarily through 'grounding': practices of embodiment and the social and cognitive normalisation of their experience. Our findings suggest that psychedelic experiences act as pivotal mental states that can facilitate transformative learning processes, challenging and expanding the ways individuals make meaning. This research contributes to the growing field of psychedelic integration by exploring the complex pathways through which people reestablish coherence and grow following ontologically challenging psychedelic experiences.

**Data availability statement:** The ethical approval from the University of Greenwich Ethics Committee (researchethics@greenwich.ac.uk) precludes sharing of the data. The full interview transcripts, although anonymised, contain potentially identifying information. Available data relevant to this study is presented in the manuscript text and in Supporting Information.

**Funding:** JE received funding from the William G Nash Foundation, Emergence Benefactors, and the Sarlo Family. EKA received funding from the Economic and Social Research Council (2582332) and Emergence Benefactors.

**Competing interests:** The authors have declared that no competing interests exist.

## Introduction

The resurgence of psychedelic research in the last two decades has unveiled a complex landscape of potential benefits and challenges associated with the use of psychedelic substances in recreational and clinical contexts. Evidence for their efficacy in the treatment of mental health disorders is growing [1] while their therapeutic potential appears to be transdiagnostic [2]. Concurrently, it is increasingly recognized that psychedelics can occasion profound shifts in metaphysical beliefs, propelling individuals into realms of consciousness that challenge their foundational worldviews. For example, multiple studies have identified a tendency following psychedelic use to shift away from physicalism, the belief that everything can be explained through physical processes, towards non-physicalist beliefs, which propose that aspects of reality may exist beyond the purely physical realm [3,4]. Single psychedelic experiences have also been shown to alter beliefs regarding the attribution of consciousness to both living and non-living entities [5]. A retrospective self-report survey by Davis and colleagues' [6] revealed that half of all atheists who had *N,N*-dimethyltryptamine (DMT) experiences reported a transition to non-atheism.

Such ontological shifts (concerning the fundamental nature of reality and being), particularly in contexts like the treatment of psychiatric comorbidities in late-stage cancer patients, have been associated with reductions in existential distress and death anxiety, enhancing life meaning and purpose, thereby mediating associated depression and anxiety [7–13]. It has also been theorised that psychedelic experiences promote openness and catalyse changes in meaning-making [14], offering a potential for both therapeutic [15,16] and non-therapeutic benefit [17,18]. These shifts can create a liminal space—a threshold state of ambiguity and disorientation where individuals are distanced from prior certainties. In this liminal space, individuals may experience profound transformation as they confront and reconstruct new frameworks for understanding their existence [19,20].

Several pharmacological, neural and psychological mechanisms have been proposed in attempts to answer how psychedelics affect the human mind [21]. With respect to beliefs, the (Relaxed Beliefs Under Psychedelics) REBUS model posits that psychedelics relax the brain's rigid high-level beliefs (priors) in its predictive processing, allowing sensory information to influence perception more freely [22]. This reduced dominance of top-down beliefs enhances the brain's plasticity, encouraging new interpretations and associations. Integrated Information Theory (IIT) further suggests that psychedelics increase the brain's capacity for integrating information, enhancing conscious experiences through more complex, interconnected neural activity [23,24]. Together, these frameworks suggest that psychedelic-induced shifts in hierarchical processing and the integration of information can lead to richer and more expansive cognitive states, often perceived as profound or insightful.

The potential for profound changes in beliefs and identity following psychedelic experiences raises ethical dilemmas [25,26]. Given the relative unpredictability of such transformations, the extent to which informed consent to this process of change can be reliably attained ahead of psychedelic use remains a contentious issue [27,28]. This concern is compounded by the fact that a significant minority of

psychedelic experiences, particularly acutely challenging ones, may lead to long-term psychological distress [29–34]. In a study of 608 individuals who reported extended difficulties after the use of psychedelics, a third reported struggling longer than a year [35] and 50% reported protracted ontological challenges, defined as the way they understood reality and existence. The most common theme in the qualitative analysis of these challenges was 'existential struggle' which related to meaning making, including expressions of existential confusion and struggles to make sense of psychedelic experiences and reality.

The existential confusion reported by psychedelic users—where long-held beliefs about the nature of reality and self are called into question—can be interpreted in neural terms as a temporary destabilization of the brain's usual mechanisms for organizing experience. Psychedelics disrupt the brain's default mode network (DMN) involved in maintaining a coherent sense of self [36] and this disruption can lead to a reduction in the integration of information that maintains the sense of a distinct, stable ego [23]. As a result, users may experience a breakdown of the usual distinctions between self and world. Enactive cognitive scientists challenge the causally linear assumptions of these theories as reductionist and argue for the need to consider the principles of autonomy and dynamic co-emergence to the psychedelic brain-experience relationship (see [37] for a discussion on this tension). Ultimately, brain process-based theories provide useful insight on neural mechanisms but do not fully account for individuals' appraisals (meaning assignment) and narrativization of their experience in influencing its outcome.

Crucially, the contrasting effects of psychedelics suggest that they are neither inherently therapeutic, nor harmful, but elicitors of pivotal mental states (PiMS), '*transient, intense hyper-plastic mind and brain states, with exceptional potential for mediating psychological transformation*' [[38], p.320]. Just as traumatic experiences can lead to posttraumatic stress disorder (PTSD) or post-traumatic growth (PTG) [39], these mental states of potentiality are outcome-agnostic, influenced heavily by the individual's capacity and resources for meaning making.

The process of making sense of and learning from psychedelic experiences, particularly those that are challenging, is commonly described as 'integration' [40,41]. Integration, as theorized by Siegel in a broader psychological context, involves "linking differentiated experiences" to create coherence in one's mental and emotional landscape [42]. This involves connecting isolated or disruptive mental states to form an interconnected whole, which is essential for fostering mental health, resilience, and self-regulation. In psychedelic contexts, this process is crucial, as individuals attempt to weave profound and often disorienting experiences into their everyday lives. Effective integration may lead to psychological growth, where individuals find new perspectives and flexibility. Conversely, disruptions to this process may lead to trauma, where experiences remain fragmented and distressing.

Psychedelic integration encompasses a broad spectrum of approaches and practices, ranging from psychotherapy to personal rituals like meditation, yoga, self-education and engagement with supportive communities [43,44]. Despite its critical role in the psychedelic journey, the nuances of the integration process remain poorly understood, with scant empirical data to guide effective practices [45–47]. Interestingly, some studies suggest that working through the challenges and difficulties that may arise during integration, such as initial confusion or emotional distress, can ultimately lead to positive psychological outcomes [48,49], further highlighting the complexity of the integration process.

In addition to the lack of systematic research exploring the efficacy of different psychotherapeutic approaches to integration, it is now known that long-term psychological difficulties following psychedelic use manifest in a range of ways, and yet there is a dearth of research exploring the integration of specific difficulty types [44,50]. For instance, there may be very different approaches needed with the integration of long-term perceptual changes, such as acquired synaesthesia [51] or Hallucinogen Persisting Perception Disorder (HPPD), which is related to elevated anxiety [52], or the extended psychological distress associated with the fairly prevalent incidence of ontological and existential challenges.

Existential challenges arising through the use of psychedelics often involve intense experiences of fear of insanity, death, and loss of control during the trip [54]. Stanislav Grof described these as "no exit" situations, where individuals feel trapped within overwhelming existential themes [53]. These themes align with Yalom's core existential concerns of death,

freedom (responsibility), isolation, and meaninglessness [54,55]. Grof suggested that if these existential themes arise during the acute psychedelic experience and are not adequately processed or integrated, they may continue to affect the individual after the session, potentially manifesting as anxiety or 'flashbacks'. However, in some cases, existential distress may occur not because of a direct encounter with meaninglessness, but rather due to an overwhelming sense of meaning in comparison to previous beliefs and worldviews, a phenomenon that may be better understood as 'ontological shock'.

Originally proposed by theologian Paul Tillich [56] ontological shock describes the profound disorientation that occurs when individuals confront the 'threat of non-being' – an awareness of mortality and the potential loss of one's identity or meaning in life. This term has since been applied to a wide range of profound worldview and metaphysical belief challenges such as those arising from exceptional experiences, including paranormal-like experiences [58], alien-abduction-like experiences [57], DMT or near-death experiences [60] and other psychedelic experiences, such as becoming another species or encountering entities [58,59]. According to R.D. Laing [60], experiences that create profound cognitive dissonance, between prior perceptions of reality and those arising following an anomalous experience that appears genuine, can threaten 'ontological security' – the sense of stability in one's understanding of reality that creates a feeling of trust in the world.

These challenges to ontological security can offer opportunities for growth, particularly in psychological flexibility, which has been identified as a mediator of therapeutic outcomes in psychedelic research [61]. If integrated within a non-judgemental and compassionate context these experiences may foster positive meaning-making and transformation [57]. However, if left unresolved or unsupported they can lead to crises that amplify distress [62,63].

### Aims

The objective of this retrospective qualitative study was to conduct an in-depth exploration of self-reported experiences of existential distress following the use of psychedelic substances. The study aimed to identify common themes among individuals who reported such difficulties after a past psychedelic experience, in order to enhance the understanding of this under-investigated phenomenon and its effective management. The following research questions were addressed:

1. What is the phenomenology of the extended difficulties experienced by individuals who dealt with existential struggle following their psychedelic experiences?

2. What do these individuals find helpful and unhelpful in managing their difficulties?

3. How do these individuals integrate their experiences and what learnings do they take forward?

## Method

### Design

This study forms part of the Challenging Psychedelic Experiences Project, which is a mixed-methods, multi-country investigation into extended difficulties following the use of psychedelics. As of early 2024, the project has involved an online survey of 608 individuals who report difficulties lasting more than a day following a psychedelic experience [35,44], plus the qualitative study reported here. For the current study, a select group of individuals from the survey sample were interviewed who showed existential-type difficulties, and these were submitted to a thematic analysis.

### Participants

Participants were recruited through various channels, including multiple social media platforms, a newsletter focused on psychology and philosophy, email lists targeting students, and a newspaper advertisement. There were no financial incentives for participation. There were three criteria for participation: (1) be aged 18 or over, (2) be proficient or fluent in

English, and (3) To have experienced difficulties after taking a psychedelic drug that negatively impacted functioning for more than a day afterwards. Those participants who offered to participate in a follow-up interview, and provided an email address, were collated into a list for interview recruitment.

Participants were selected for this study based on the presence of a specific theme, "Existential Struggle," in their responses to open-ended survey questions. This theme, identified through qualitative analysis of the survey data, was reported by 102 participants (17% of the total survey sample). The theme subsumed descriptions of struggling with and being confused by (a) a sense of reality having shifted or changed, (b) making sense of experience, with (c) lost or changed meaning and beliefs, the weeks and months following a psychedelic experience. The rationale for selecting participants exhibiting this specific theme was to conduct an in-depth exploration of the challenges associated with this particular kind of difficulty, and its potential relationship to related concepts such as 'ontological shock'.

Initially, 28 individuals participated in the study; however, two transcripts were subsequently excluded from the analysis as they did not directly address the study's aim of exploring existential difficulties following psychedelic experiences. The decision to conclude recruitment at a sample size of 26 was reached collaboratively by the research team, based on the attainment of data saturation in the emerging analytical themes [64].

Table 1 presents the demographic characteristics of the participants and provides information regarding the substance and context of the psychedelic experience discussed during the interviews. Where an age range is given in the table this is in absence of exact ages reported to the interviewer and is sourced from participants' original surveys. The substance(s) taken column shows the substances taken prior to the experiences in question (the ontologically challenging psychedelic experiences that led to extended ontological difficulties). Multiple substances indicate poly-substance use within a trip and 'different trips' indicate that the participant referred to more than one experience that led to these difficulties.

We included both classic psychedelics (e.g., psilocybin, LSD) and non-typical psychedelics, (e.g., MDMA and cannabis) as our primary focus was on the phenomenology of participants' subjective experiences that they defined as psychedelic rather than the specific neural mechanisms of each substance [65].

## Procedure and data collection

Semi-structured, one-to-one online interviews were conducted between August and November 2023, lasting between 40 and 95 minutes. The interviews were conducted on Zoom or Teams by one of six researchers, all of whom had prior experience with qualitative research interviewing. The interview topic guide was developed based on existing theory and research concerning difficulties associated with psychedelic experiences. It covered the following topics: (1) circumstances of the psychedelic experience (age, years ago, substance(s), life situation, intentions and expectations; (2) the experience itself (people involved, duration, experiences during the trip); (3) extended difficulties (kinds of difficulty, duration, effects, beliefs, interactions with others); (4) management of difficulties (coping strategies employed, helpful and unhelpful factors); (6) insights on integration, change and support; (7) suggestions for supporting others in the future. There was no incentive offered for participation. Interviews were audio-recorded and then transcribed using Otter.ai, followed by a thorough transcript review and comparison against the audio by each interviewer to ensure accuracy.

## Data analysis

Three members of the research team (EKA, JE, PM) carried out a thematic analysis of interviews, with a predominantly inductive approach to theme creation. One research team member (OR) acted as an additional consultant during specific phases of the analysis. Themes and subthemes were regularly discussed between the three researchers throughout the analysis stage to ensure appropriate categorisation of codes, ensuring that theme development was consensual and

**Table 1. Demographic and psychedelic experience information for participants.**

| Participant Pseudonym | Gender | Ethnicity | Nationality | Education level | Age at psychedelic experience | Age at interview | Substance/s taken | Setting of experience |
|---|---|---|---|---|---|---|---|---|
| Aaron | Male | White | USA | Bachelors degree | Unknown | 45-55 | Psilocybin, ketamine and MDMA (different trips) | With 'journey partner' facilitator |
| Georgia | Female | White | South Africa | Bachelors degree | 55 | 56 | Psilocybin | At home alone |
| Will | Male | White | UK | Masters degree | 26 | 34 | LSD | At home with partner |
| Emine | Female | White | UK | Not specified | 42 | 48 | Psilocybin | Guided group underground ceremony |
| Kirsty | Female | White | USA | Masters degree | 43 | 44 | Psilocybin | Guided group retreat in the Netherlands |
| Beth | Female | White | UK | Masters degree | 24 | 27 | LSD, nitrous oxide, cannabis | BnB with partner |
| Caitlin | Female | White | USA | PhD | 28 | 35 | Ayahuasca | Peru small group, ceremony, couple guides |
| Noah | Male | White | Israel | Bachelors degree | 31 | 35 | LSD | Party |
| Jessie | Female | White | USA | Masters degree | 35 | 36 | Hapé | Spiritual women's group retreat |
| Max | Male | White | USA | Masters degree | 21 | 27 | Psilocybin | With two friends in Colorado mountains |
| Cal | Non-binary | Ashkenazi | USA | Masters degree | 17 | unknown | DMT, cannabis, | Party |
| Don | Male | White | USA | Masters degree | 50s | 50s | Cannabis | With a friend at his home |
| Adrienne | Female | White | UK/France | Not specified | 37 | 39 | Psilocybin | Alone in her apartment |
| Chris | Male | White | UK | Bachelors degree | 27 | 34 | Psilocybin | Retreat centre in Holland |
| Harry | Male | White | UK | Masters degree | 22 | 25 | Psilocybin | Alone at university |
| Steve | Male | White | UK | Bachelors degree | 22 | 23 | Psilocybin | Alone in his apartment |
| Jack | Male | White | Belgium | Masters degree | 23 | 27 | Psilocybin | Festival |
| Clara | Female | White | UK | Masters degree | 34 | 41 | Changa DMT, MDMA and psilocybin | With ceremony community (not in ceremony) |
| Teri | Female | White | USA | Bachelors degree | 36 | 36 | Psilocybin | Beach, alone |
| Theo | Male | White | UK | Bachelors degree | 26 | 33 | Ayahuasca | Ceremony with Ecuadorian shaman |
| Elijah | Male | White | UK | Bachelors degree | 24 | 30 | LSD | Friends house |
| Fred | Male | White | Germany | PhD | 32 | 33 | DMT | At home alone |
| Youssef | Male | Middle Eastern | Lebanon | Bachelors degree | 25-34 | 25-34 | Ayahuasca | Retreat centre, with Amazonian shaman |
| Cora | Female | White | USA | Masters degree | 25-34 | 25-34 | Cannabis | In nature, in India, with trusted friends |
| Jan | Male | White | South Africa | High School | 21 | 32 | LSD | Festival |
| Ida | Female | White | Belgium | Masters degree | 27 | 29 | MDMA | Festival |

collaborative. We followed the six stages of thematic analysis as outlined in Braun and Clarke's [66] guide ('Familiarising yourself with the dataset; Generating initial codes; Searching for themes; Reviewing themes; Defining and naming themes; Producing the report'). All interviews were analysed individually prior to cross-case themes being developed, and codes were collated on a Miro online whiteboard for visualising code clusters that could be developed into themes.

One difference from Braun and Clarke's approach to reporting a thematic analysis and the method employed within this study, is that we report the frequency of themes, rather than using less precise quantifiers such as "many participants" or "some participants". We do so because we are of the view that it enhances transparency and rigour of reporting to report the more accurate information [67]; all frequencies and percentages appear in the table of themes and subthemes in Supplementary Materials.

### Reflexive approach

The Challenging Psychedelic Experiences Project aims to contribute towards building a safer culture by studying and communicating psychedelic-induced difficulties and what helps overcome them. As researchers on this study, we are exploring the complex impacts of ontologically challenging psychedelic experiences. 9 authors are based in the UK, 1 in the U.S. and 1 in Costa Rica. Our overriding approach as a group is to explore the heterogeneity of psychedelic experience. 11 authors have had psychedelic experiences and 8 have dealt with extended difficulties that related to subsequent ontological challenges. We recognise that our shared familiarity with the subject matter allows us to approach participants' perspectives with empathy and understanding, while remaining aware of potential biases that such familiarity may introduce to our analysis. This reflexive awareness has guided us to maintain rigor and openness in our interpretations through the research process and in our communication, as we worked to ensure our findings are grounded in the lived experiences shared by the study participants.

### Research ethics

Ethical approval for the study and research procedure was obtained from the University of Greenwich Research Ethics Board (Project ID: 21.5.7.20). We followed ethical guidelines set out by the British Psychological Society for conducting qualitative research. We sought written informed consent from participants, and participants were given the right to withdraw up to 2 weeks following the interview. All data were held securely and confidentially. Transcripts were fully anonymised so that names, places and other possible identifying details were removed.

## Results

Themes identified through the thematic analysis are described below and arranged within 11 sections. Theme names are shown in bold, and subtheme names are shown in italics.

To enhance the conciseness of the analytical presentation, illustrative quotes are only presented for themes that relate specifically to the principle aim, i.e., elucidating the phenomenology of existential and ontological difficulties that emerge after a psychedelic experience, and for strategies to cope with such difficulties. A full list of themes with frequencies, percentages and indicative quotes is provided in supplementary material.

### Life contexts

Participants described **transitional, exploratory and turbulent life circumstances** prior to their challenging psychedelic experience, which may have contributed to their 'set' (mindset and mental state) at the time. A *period of healing and self-exploration* was described by 9 participants. Twelve others were in a *highly transitional time*, such as changing jobs, on the cusp of major changes and wanted "to shake everything up". Twelve were going through a t*urbulent or stressful time*. Others described *torment, turmoil and instability*. Only two participants reported no significant personal difficulties prior to their experience.

### A varied range of intentions

Participants described **Varied intentions** for their psychedelic experience. Several were encouraged by positive associations they had made with psychedelics. Five were spurred on by the precedent of having *Positive prior experiences*. Being inspired by *Psychedelic research and media hype* was mentioned by three. *Exploration or curiosity* was the intention described by nine people. Two mentioned *Seeking a spiritual experience.* Seven participants had the intention to *Have fun. Self-development* intentions were relayed by 5 participants. An active *Search for Healing* was iterated by 6 people. Five others articulated intentions to help gain *Clarity on life decisions*, such as with moving, jobs, or relationships. One participant, Cal, was subject to the experience accidentally, and thus without an intention, by unwittingly smoking cannabis laced with DMT.

### The acute psychedelic experience

The acute psychedelic experiences that led to the extended difficulties were described in broadly negative terms, with twelve participants explicitly referring to their experience as traumatic. 16 participants conveyed the theme **Existential concern.** For 9 individuals, this existential concern involved *Perceived physical or ego death.* The theme *Overwhelming responsibility* was conveyed by 7 of the participants. Experiences echoing a sense of *Meaninglessness or emptiness* were shared by 5 individuals. The final theme under this existential umbrella referred to *Solipsistic isolation or Aloneness,* reported by 6 participants.

A feeling of there being *No Exit* was experienced by 14 participants. In at least two, this expressed itself as a *Fear of dying. Fear of insanity or permanent damage* was variously expressed by 6 subjects as having "actually lost my mind" (Georgia), "I think I broke my brain" (Max), or "I'm just always going to be this way" (Cal), powerfully rendered by Caitlin remarking "I have done something, like, *deeply* irreparable to my existence." 12 subjects reported severe states of uncertainty embedded in states of *Confusion.* Seven expressed an *Inability to compute, or articulate the experience,* and six said that they *Received a confusing message,* which may have been a conveyance via the experience itself or from a specific 'other' agent, and in 2 cases related to a questioning of their sexual orientation.

The second theme **Ontologically Challenging Experiences** (OCEs) was conveyed by 17 participants. This included the subtheme of *Challenging mystical or religious experiences.* expressed by 8 interviewees. A *perception of an entity encounter or possession* was shared by six participants. Seven participants related an experience of being *Given insights or gnosis.* Six interviewees shared experiences of glimpsing a *Hellish afterlife.* Four participants' experience encompassed *Extra-sensory perception & Out-of-body experiences*. Other OCEs involved 2 individuals reporting both *Time travel* and *Transformations.* Elijah related the "pure chaos" of becoming different people through history as well as objects upon observing them, and seeing friends convert into angels and devils.

The theme **Expressions of fear** was conveyed by 16 people, with sub themes as follows. *Paranoia* was felt by seven, such as deep distrust or that "everyone in the world knew my deepest secrets" (Will). *Feeling unsafe* was shared by four, typified by Clara's non-consensual sexual encounter with members of the 'ayahuasca familia', a cult-like community and example of "unheld psychedelic groups [which] are incredibly dangerous". *Anxiety or terror* was communicated explicitly by 11 individuals.

**Other aspects of the acute phenomenology** included the subtheme *Overwhelm,* mentioned by 8 with references to sensory overload. *Aggressive reactions* were offered by 2, such as Aaron who was "trying to provoke something that felt like it wasn't 'coming from within me' via violent acts.

Finally, *13* participants described **exiting their trip** feeling *shaken*, *fragile and vulnerable*; *anxious and overwhelmed*; or *physically exhausted.* Georgia notes she felt *'physically as if [she] had been in a battle.'*

### Worldview shifts

Interviewees reported experiencing major **worldview shifts** following their psychedelic experiences, which sometimes took years and were often bewildering to go through. For example, Adrienne started off the COVID-19 pandemic as an atheist dominatrix and, after an extremely challenging psychedelic experience, ended the pandemic by taking vows to

become a Buddhist nun. Don transitioned from being an atheist US Airforce clerk to becoming a medium and astral traveller in a channelling community.

The most common shift, experienced by eight of the 26 interviewees, was from a materialist-atheist to a spiritual worldview:

*I think the one big, big, big issue of this all was actually that I didn't have a spiritual framework to place this experience in. [I became] less focused on this purely scientific materialistic worldview somehow. This experience just kind of cracked it open. (Fred)*

For four interviewees, the belief-shift involved a loss of faith in their previous idea of God and a *move away from traditional theocentric religion to a more spiritual, mystical or magical worldview*:

*My relationship with spirituality absolutely changed because at that time in my life, I was considering becoming a rabbi and I became a pagan…I think the fact that no other power came down to help me in this huge time of need may have been part of the shift [from Judaism to becoming a Wicca priestess]. If I want change to happen, I have to do it. Which of course shifted me away from going to be a rabbi and [towards] becoming a priestess. (Cal)*

Two shifted from a spiritual seeker worldview towards a more *evidence-based scientific or sceptical worldview* as a way out of their existential crisis:

*I've written a lot about natural science. And I'm just basically trying to reconstruct a worldview that's in line with reality. I'm trying to try to stay as close to what we actually know as possible, rather than deal with these kinds of things that are all the way over there. (Steve)*

And for four interviewees, the challenging psychedelic experience ended up *undermining their faith in psychedelics*, which had previously held a central space in their spirituality.

*But going from a position where I felt that I could trust this substance almost, or that it would always work out well for me when I did this substance. It had been a guiding light. And then suddenly, something had changed. (Harry)*

## Extended difficulties

All interviewees were sampled for the study due to mentioning experiences of existential struggle and confusion after a psychedelic trip, in a prior survey. Fig 1. presents the kinds of difficulties organised into themes and the following section considers the nature of participants' difficulties in more detail.

A diagrammatic representation of extended difficulties themes and subthemes.

All interviewees experienced some implication of '**Ontological shock and existential confusion**'. For ten participants, the difficulty manifested as *Existential crisis and despair*, for example:

*Most days I would feel anxiety, fear, sit in disorientation and deeply saddened by my existence, which progressed into existential crisis… the questioning of this whole universe, why are we here, what's the point of this. (Emine)*

As an overlapping theme, eight participants articulated feelings of *Emptiness, meaninglessness and nihilism*:

*There was a lot of nihilism as well. Those extreme levels of annihilation of not wanting to do anything. What's the point of showering now? We're all gonna die anyway. (Emine)*

 

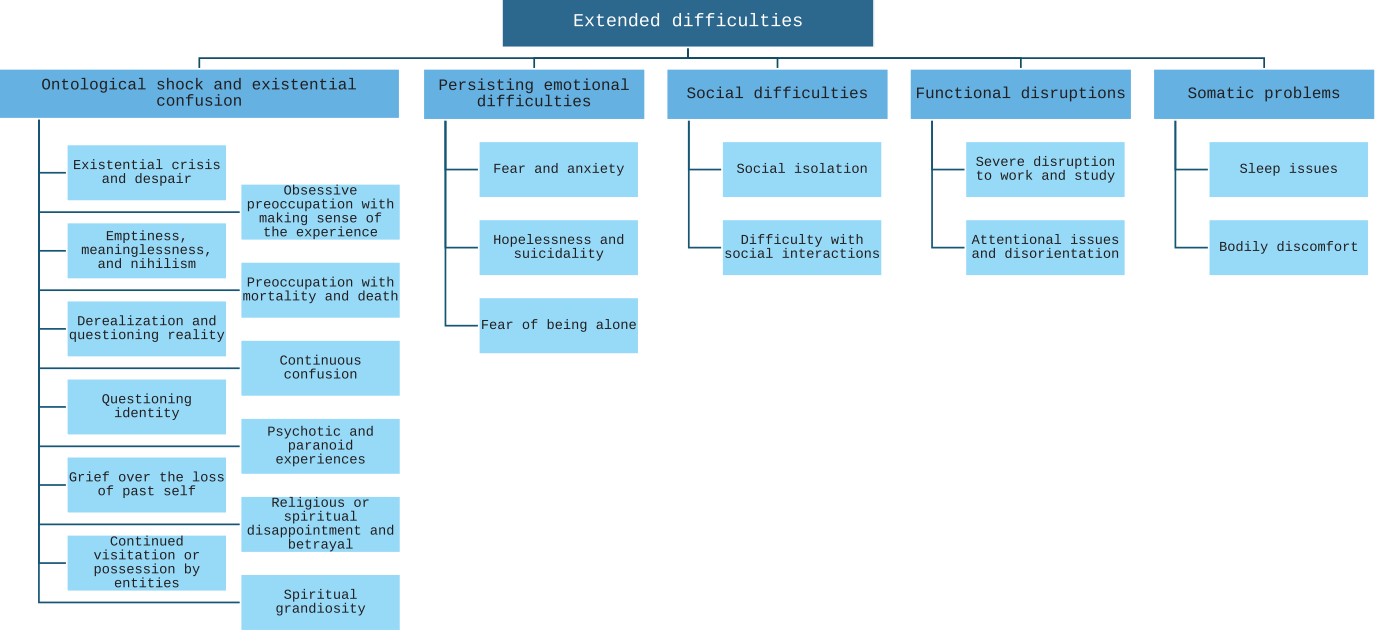

**Fig 1. Extended difficulties themes.**

**Experiences of derealization and questioning reality** were reported by 15 participants

*It felt like I was in a golden state of derealization. Yeah, dissociation but also sometimes very associated, like being very much in the moment, not being able to think about what's going to happen the next minute... It's hard to put into words... It felt as if I was a ghost somehow... I was there but also I was not there. (Ida)*

Eight more described how their challenging experience led to struggles with *Questioning their identity'*

*My sense of self, I mean, it was gone, you know. It was like it had been obliterated into a million pieces. And I couldn't really work out who I was. I just had this, like, routine of going to work. And I just stuck to that as a means of like, keeping some sense of inner structure by having some outer structure. And, like, if I'm honest, if I just, if I'd sat and thought about that for a while, for the first few weeks, I think I would have just cried because like, it didn't seem very clear who I was anymore. (Theo)*

Related to this identity shift, eight experienced *Grief over the loss of their past self*

*I would really love to somehow get back to how I used to be... [Back to] when I didn't used to think and have all this knowledge. That sort of ignorance is bliss, sort of mindset... there's a sense of loss for like the person I was because yeah, it's just gone. (Elijah)*

Six interviewees experienced encounters with perceived entities during the psychedelic experience and *Continued to feel visited or even possessed*, in a way that was both disturbing and existentially challenging:

*And I would perceive it as negative energy that I was absorbing and could not access my breath anymore to try to clear my nervous system, clear my energy body... eventually there were audio hallucinations. And so I felt as though they were out to get me for some reason, like these entities from others were out to get me. (Cora)*

Twelve participants reported *Obsessive preoccupation with making sense of their experience*, finding themselves locked into rumination:

> *I could not stop thinking about what I had seen...I was just like, couldn't focus on anything like that. It was just going round and round in my head like that. (Beth)*

Five interviewees became *Preoccupied with mortality and death*:

> *I have this feeling of like, instead of being relieved from death anxiety I was now given a death anxiety that I never had prior to having the psilocybin experience. (Max)*

Eleven participants reported *Continuous confusion* as they struggled to accommodate their ontological shock, for example:

> *When all those boundaries get confused, or you're just living in an entirely fabricated LSD dreamland, then you're like, Oh whoa, that's the world over there, like, there it is, it's gone crazy, what's happening to you? What's going on with me? All was not as it seems, and sensory data is not the real world (Will)*

Seven interviewees reported *Psychotic and paranoid* experiences

> *There were voices, many voices in my head that were psychotic, thoughts that were just so ridiculous. So twisted, so dark, so, so weird, disgusting, fearful, all the crazy mix of strange thoughts over the simplest of things (Jan)*

> *I was also slightly convinced that like the CIA would come because I thought this, you know, this information is too too esoteric for me to have access to (Beth)*

Seven interviewees experienced *Religious or spiritual disappointment and betrayal*:

> *I went looking for God and for love and for connection and I got the exact opposite. gave me this feeling of existential betrayal that to this day, I feel has created a very, I mean, profoundly distrust for me, of a very deep sense of like not feeling safe in life. (Max)*

Two participants described how they developed a sense of *Spiritual grandiosity*:

> *I had some grandiosity around it like 'I am the shit basically', like I can maintain the state of bliss, bliss, like reality is just giving me everything I want....I was living in like, a God Realm. Everything was flowing beautifully in my life...I had kind of entangled my identity with that as well as being like, spiritual superior. But...then I started losing stability. (Cora)*

Seventeen participants described **persisting emotional difficulties**

This included twelve participants struggling with 'fear and anxiety' which often involved physical anxiety and panic attacks

> *I felt like, like I can't breathe. Like I don't have any air to breathe… very extreme restlessness. Like I have to do something with myself and if I don't…I'm going to die. (Noah)*

Five participants experienced *Hopelessness and suicidality*. Four participants specifically mentioned a new and persisting *Fear of being alone*:

> *I had real difficulty with staying alone. The anxiety was overpowering. (Noah); Like somehow being on my own suddenly felt more scary or anxious. (Kirsty)*

Sixteen participants experienced **social difficulties**: Fourteen described entering a *Social isolation* exacerbated by feeling unable to talk to anyone about their experience

*It kind of just took me into this very inward hibernation. I just didn't want to be around people or, you know, talk to any-one and just worked through that. (Jessie)*

Six interviewees described *Difficulty with social interactions* due to feeling hypersensitive or paranoid

*I had the feeling that all my scales were open, like everything could just enter…. I was absorbing everything from every-one around me... I was being very anxious about my relationships. (Ida)*

*I would be sitting with people and thoughts would be rushing in my head that were basically delusions. 'What are these people thinking about me?' (Youssef)*

Twelve participants reported **functional disruptions**, including cognitive and related behavioural difficulties that affected their functioning on everyday activities. Eleven described *Severe disruption to their ability to work and study*. Four specifically referred to *Attentional issues and disorientation*

*I couldn't focus. I was a very avid reader, I could read 100 pages a day easily and would want to read more and stop. After that ceremony, I couldn't get through five pages without physically being in pain and fidgeting and wanting to leave. (Youssef)*

Ten participants described **somatic** problems, including 'sleep issues' such as nightmares described by seven; 'Sleeping was returning to hell.' (Aaron) and bodily discomfort by four:

*I was so uncomfortable in my skin. I was crawling out of my skin. I never felt more uncomfortable in my body and like, scared in this way of like, my body doesn't feel like mine (Caitlin)*

**Trauma and PTSD similarities**

Twelve participants explicitly referred to their experience as **traumatic** (i.e., 'it was really traumatising on me and my nervous system (Cora); it was absolutely traumatic' (Adrienne)). There was also significant overlap in participants' reports of their lasting difficulties with the DSM-5 criteria for Post Traumatic Stress Disorder (PTSD), namely: re-experiencing, avoidance of thoughts of trauma, negative alterations in cognitions and mood, alterations in arousal and reactivity and significant functional impairment that lasted over one month.

Fifteen participants reported **flashbacks** to the trauma of the experience **and other forms of re-experiencing**.

*For the first six months after that… like maybe every three to four weeks flashes of terror... I was having a visual effect of the world starting to dissolve and it was like a flashback - whatever you want to say, a recurrence of the journey (Aaron)*

**Triggers** that participants identified as bringing them back to the mental space of their psychedelic experience included: *meditation and breathwork*, *substance use* (including further psychedelic trips), *discussions related to psyche-delics or the content of their trip*, and *sleep-related triggers* (including falling asleep and nightmares).

**What helped manage the difficulties?**

A range of support and coping practices were reported as helpful in alleviating extended difficulties, which are visually summarised in Fig 2.

Themes and subthemes of what participants found helpful for managing their difficulties.

When asked about the kinds of practices they found helpful in navigating their difficulties and why, almost all (22 out of 26) participants made mention of **practices that helped them 'ground' themselves** and let go of cognitive preoccupations.

*I really started to get into… body oriented practices to ground because that was in first instance what I really needed, to ground. To be present again… just like with the sensations in my body, and, learning to recognise, if my stomach contracts, what does this mean? Just letting it be? Not thinking, Oh, my God, I feel awful and this is never going to end and then it was very cognitive again. Learning to feel into the moment, into my body again. (Ida)*

*Any like, hands on like touch, like grounding kind of experience was helpful (Caitlin)*

*[I felt like] I need to get back in my body. I need to engage with this physical thing to somehow learn: Where are we? (Fred)*

Specific practices that were noted for their grounding effects were *yoga and other body work* (including acupuncture, massage, breathwork), *trauma release exercises* (including shaking and screaming), *experiences with water* (hot baths, cold showers, swimming, surfing) and *spending time in nature or with animals*.

*I felt as if I was kind of outside my body, so just doing things like just stretching, yoga, and then TRE, which is trauma release. That's where you get your body into a state and your legs shake and that's, they use it for releasing trauma... it was like 'There's trauma in my body and I've got to get rid of it.... I've got to shake, I've got to get this stuff out of my*

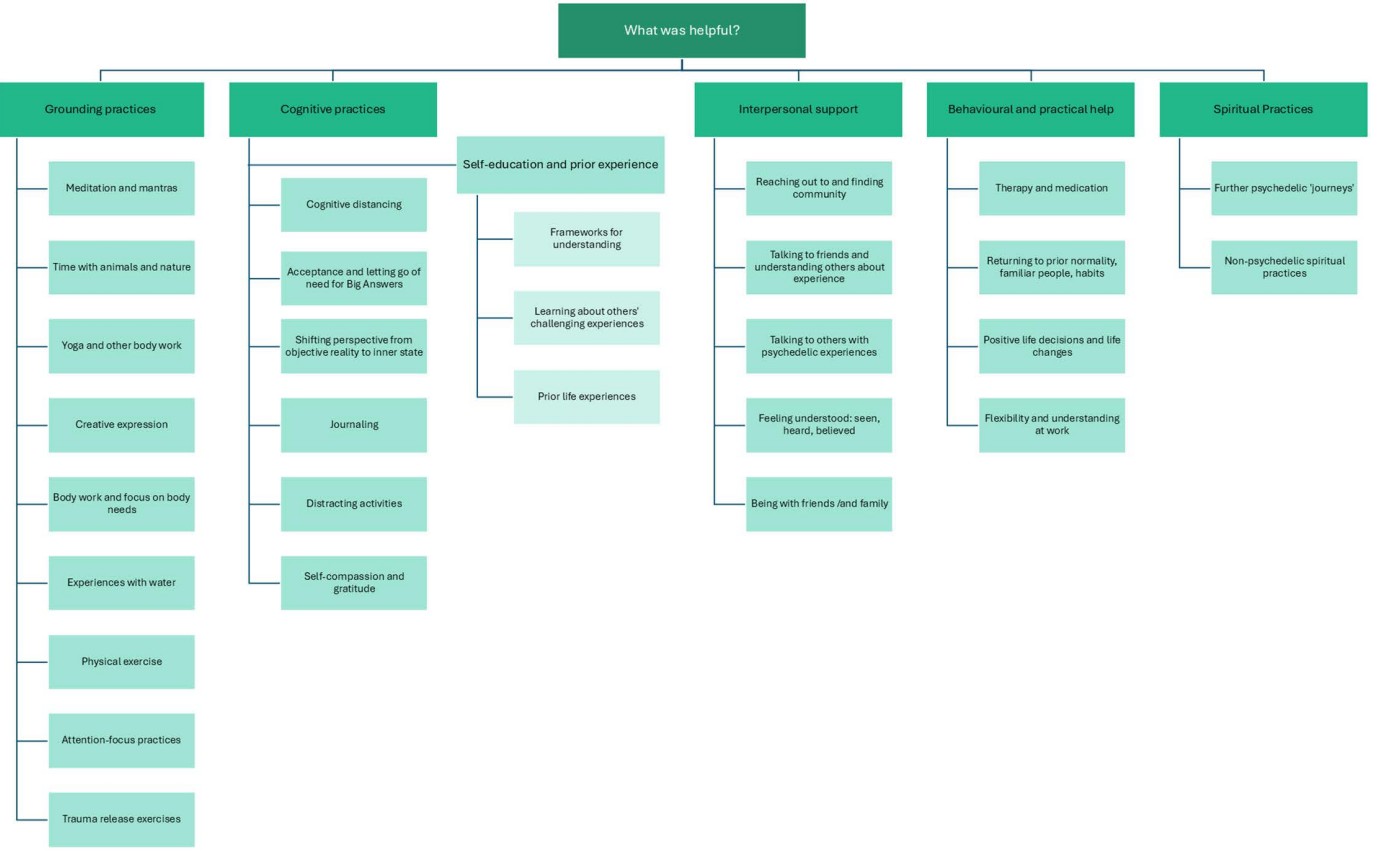

**Fig 2. Helpful practices and support.**

*body'. And then things like just walking barefoot. So you know luckily, I know about grounding… I just felt like I needed to have my feet on the grass and that sort of thing. (Georgia)*

*Creative expression* (including drawing, writing, singing and playing musical instruments) was also reported as helpful.

*I picked up the guitar. After a year and so from playing the guitar, I can tell you that this practice saved my life… It… puts all the worries, all of the panic, all of the stress away… The world just drops. It's just me and the guitar, me singing. My heart soars, my mind quiets down. I feel grounded. (Youssef)*

While some participants noted that *distracting activities* (work, tv-series, audiobooks) helped them cope, *attention-focus practices* (including mantras, nail-art, colouring, counting and certain types of meditation) were also found to be helpful.

*One of the things that I did was a simple countdown practice. So…when I was getting into obsessive thinking or high anxiety regarding the experience, I would count down from three…So not fighting against the desire to anxiously think or ruminate…but clearly labelling myself, this is here, but this isn't helping. Then I would redirect my attention back to my body into the present moment and just try to be with my life as it was happening in the moment. (Max)*

*Meditation* was reported as both helpful and unhelpful by different participants and was also identified as a trigger back to the psychedelic experience. A quote by Ida addresses these conflicting functions of meditation:

*I think with meditation, it's really important to find a sort of balance [with] turning inwards and doing introspection, I was too caught up into that and then I was not experiencing the world outside of me. Because I was so closed in my thoughts, I didn't understand anything. So I kept closed in my head, I got stuck in my head. But then, from that point on, I started to go outward again. Trying to experience what am I feeling? What am I hearing? What do I see? And just objectively naming it, trying to not judge things, just try to observe. And, yeah, these are both meditative practices, but... one is going inward, the other one is going outward, and I really needed to go outwards.*

**Cognitive reframing and distancing practices** were reported as helpful by 17 participants. This included the subtheme of *cognitive distancing:*

*It's a distancing thing. I think the version that I have since come to is, 'What is this part of me trying to tell this other part of myself? What can I learn from this?' And I find that particular framing really helpful because it both creates that mental distance to analyse and notice what's happening. (Aaron)*

Relatedly, some participants pointed out how *shifting perspective on their experience from an objective reality to an inner state* helped them:

*What I saw on my trip was just that – the quality of my mind and how it was determining the quality of my life. The trip showed me in what direction my mind was pointed. That helped me strip this sense of objectiveness from my experience, and allowed me to connect with the sense that what I experienced was not true objectively. It was simply true and relevant to me. And that's something that definitely helped stabilise me a lot… (Max)*

*Acceptance and letting go of a need for answers* was also noted, exemplified in a quote by Steve:

*The mind's always trying to grasp onto something but… the mind will never get there. It's coming to terms with infinite complexity rather than trying to figure it out...Now it's just about really making peace with the fact that maybe I won't*

*know, and it's just about really stopping myself from trying to figure things out. I was treating it as a philosophical exploration sort of thing. When really, it's quite like a visceral experience.*

A few participants found *journaling* helped them break down and process their experience:

*Journaling as well to just write the whole journey out so I could go back and I could read it… wrote probably about three or four pages, and then I would go back and just read one paragraph, and then just work on that. And then maybe the next day, go and do another. So it was all down there and then I could just deal with little bits every day as I was sort of integrating, rather than having everything here that was just this whole kind of mess of stuff (Georgia)*

Some specifically found a reframing towards *gratitude and self-compassion* helpful:

*Gratitude...just makes you a happier, better human being. But also it helped me reframe a lot of these things? Like, okay, I don't understand existence, but I'm alive. Isn't that great? Like, look at life out there. It's amazing… let's just be grateful...to be able to live... to not have died, and to [make] the most of it (Fred)*

Efforts towards *self-education and drawing from prior experience* were reported as helpful by 15 participants. In contrast to those noting they were helped from practising distancing and letting go of a search for answers, other participants noted they were helped by adopting *frameworks for understanding* their experience, including philosophical, spiritual and scientific ideas.

*I read about spiritual emergency and realised you know, there is a way out of this. You think like that's it - I've really done it this time, there's no going back again, like I'm gonna be crazy forever. So hearing that narrative that that wasn't the case was just lifesaving itself. I understood the significance of what I was going through as something that could be overcome (Beth)*

*Learning about others' challenging experience*s was also noted as helpful, as was *drawing from one's own prior life experiences*:

*Because of my age, I have done a lot of work... having more life experience and just being interested, perhaps, in things like consciousness and psychology and, so just well read on all of those subjects. And then just life in general, when you get to 56, 57, you've dealt with quite a lot of trauma anyway! (Georgia)*

**Interpersonal support** was noted as helpful by 22 participants.

This included *talking to friends and loved ones about their experiences, talking to others with psychedelic experiences* and *reaching out to and finding community*:

*I discovered a spiritual awakening sharing circle...that was literally my only support because I was going through this all alone, living alone in my flat. But every month I had this group I would go to, and for five minutes I would talk about this crazy stuff I was going through with this demon and people didn't judge me. And that was what kept me going (Adrienne)*

A few participants pointed out it was the element of *feeling understood, seen, heard and believed* that was helpful (*i.e., talking with friends, I think that helped because I felt seen and heard and held, Jessie*) and just *being with friends and family* made a difference.

**Behavioural and practical help** was reported by 16 participants. This included *therapy and medication*, *returning to prior normality* (familiar people and habits), making *positive life decisions and changes* and the importance of *flexibility and understanding at the workplace*.

Eleven participants noted they were helped by **spiritual practices**. These included *further psychedelic journeys*

*I saw... really evil, horrible, disgusting images again… But what I was dealing with there, came out it was fear. And so once again, I kind of almost had to go to hell. And then through that Ayahuasca journey, I was able to release so much fear. (Georgia).*

as well as *non-psychedelic spiritual practices*

*I then went on to have an out of body experience for about 30 seconds during that retreat… it showed me that there was plenty of challenge that can be can be undertaken without ingesting a consciousness altering drug (Theo)*

**What was unhelpful in dealing with extended difficulties?**

We asked interviewees what they found unhelpful in coping with their extended difficulties. **Lack of support and understanding from others** was described by 18 participants in the following subthemes. *Lacking an understanding community and others to speak to* impacted 10 participants, for example:

*The fact that I didn't have anyone I could speak to... not having a community before I did that was probably very unhelpful (Georgia)*

Relatedly, some interviewees described a *Lack of broader cultural understanding of psychedelic experiences*:

*You do this and you have this amazing experience and unlock these connections within, like amongst people in nature and in the world, and yet you're stuck in a life where not everybody understands that and it's very isolating (Teri)*

Eight participants found *Others' spiritual interpretations of their experience* unhelpful whether from online forums or their retreat guides themselves:

*What's so scary is that he would pin everything to an entity possession, which was even scarier. So, instead of taking ownership of this compromised facilitation, instead of taking responsibility, he would say oh, you must have been possessed by [an] entity. (Emine)*

Three participants noted how *Lacking integration support from their guide* was unhelpful

*They were very absent. I remember emailing them once and just being like,I don't know what's happening. Can you help or anything you could advise? and they were like 'Do more medicine'.. Not helpful. (Emine)*

Two participants found that *Avoiding professional help proved unhelpful.* Youssef *'didn't speak to a therapist for 3 months out of fear of being sectioned'.*

The second main theme in unhelpful coping was ***Obsessive and escapist coping*** (14 interviewees). Nine participants tried in vain to find answers in *'further substance use',* including cannabis and psychedelics *'I took a high dose again. And*

*though, psychologically, I came back down much faster, and felt like myself, the same loop came up.'* Three interviewees found an *'Obsessive search for intellectual answers'* to be unhelpful, as in the words of Max:

> *Researching existential matters made things worse…trying to intellectually solve the experience, trying to think my way out of it…to obsessively find a book or a talk or something. Some magical sentence about the nature of self and reality that is going to cure my anxiety… The search for the magic bullet, that is definitely not an effective strategy.*

**Perceived unmet needs and suggestions for the future**

Eighteen participants noted a need for **increased awareness and societal infrastructures to support difficult psychedelic experiences**. This included a need for *public awareness of potential harms of psychedelics:*

> *People need to know that psychedelics does not mean it's a quick fix. I think it's so important that you know what you're getting into. Because it's not always going to be universal love and the flowers talking to you, sometimes it's going to be going to hell (Georgia)*

but also *awareness that there is a way through the difficulties*:

> *Knowing that there is a path through and... also keep trying, don't give up. That there is something else on the other side (Kirsty).*

Nine participants noted the need for *access to support structures and* specifically for *structures that enable sharing and support*, during and after the psychedelic experience:

> *Even at festivals there's tents like PsyCare. Most festivals usually have a place where people having hardships can go, which is great... knowing that there's friendly people who are just there to help, usually in a comfy setting where you can just go and hang out or ask them help or to just be there like, it's just really reassuring. Also online versions of that could exist (Beth).*

Five participants noted that *legalisation and decriminalisation* of these substances would benefit those that struggle with their effects

> *Just not feeling like you had to be afraid of, like, who you talk to about it, like am I gonna get in trouble if I talk to the wrong person about this? So I think the legal– I actually think, like, legalising the drugs will help tremendously (Teri).*

Participants also shared their **suggestions for others based on hindsight**. These included advice for cautious use to 'start small and go slow' and well-considered preparation for the experience; coming into it with respect and caution, and having developed a toolkit of resources to draw from for support

> *To have a basket of different things that you can use… So that people would know that they have those resources, if you have a challenging psychedelic journey, these are the resources that you can try (Georgia).*

They also advised *journeying with informed others*, including experienced friends and professionals, and making sure they *dedicate time after the experience for integration*.

### Concerns about psychedelics

Twelve participants expressed **concerns about different aspects of psychedelic use and how the substances are portrayed in public narratives**. Eight participants pointed to the *Underestimation of harms from psychedelic use.* Four participants explicitly referred to *Issues* they identified *within psychedelic associated cultures* they were engaged in, pointing to spiritual narcissism and bypassing. Three participants spoke of the *Risk of codependency and abuse* by individuals who claim authority in the psychedelic space. Emine highlights the need to reinforce self-agency around psychedelic use:

> *My wish… is that we help people to become more autonomous rather than codependent even months after their sessions. And how do you do that? By continually reinforcing the self-agency and helping them see and become more aware of their own challenges in the mundane.*

Two participants also pointed to related concerns regarding the *Commercialisation and marketing of psychedelic substances.*

### Positive outcomes and learnings

Participants reported a variety of positive outcomes within their reflections of their challenges and how they overcame them. Three main themes were identified:

Firstly, **Self development and spiritual growth** was described by 14 participants. Participants described benefits of psychological maturation, e.g., *I think it spiritually and psychologically matured me very fast in a way that I imagined could have happened over you know, years or decades even (Caitlin).* They also referred to an *expanded awareness* which encompassed mindfulness and reflective thinking, e.g., *I became a lot more aware of my thoughts, more mindful (Beth)*

Others reported their experiences helped them make decisions for positive life changes (*I made decisions after that point that were just like yeah, it was just more clear what felt aligned and worth it (Caitlin)*

Four participants reported they felt empowered through overcoming their challenging experiences: *I feel massively resourced, that if I can handle that and get through it then I can kind of handle anything (Clara)*

Four participants described finding a new sense of purpose as well as feeling humbled by their experience:

> *It's a huge part of my life story... I've grown to know the importance of experiencing the full range of human emotion. And just like really accepting the fullness of being human, so I feel more human... I've been humbled in a way... being human. I'm able to see and integrate the complexity of what it is to be human and that feels really meaningful, personally, and in my work (Cora)*

Secondly, **Increased prosociality** was found in 12 participants' transcripts. Eight participants felt they had gained a *new sense of compassion, love and understanding for others*. Seven participants felt *inspired to help others* who might go through similar struggles. Four participants reflected on how their experience helped them *realise the value of connecting with others and feel gratitude* for these connections

> *It made me value again, the friendships or the family or the relationships that I have, you know, more strongly, more gratitude for that. (Jessie)*

Thirdly, **Improved mental health and full recovery** was noted by 3 participants who explicitly referred to the benefits that overcoming their challenging experience had for their mental health.

Participants reported diverse **understandings of integration**. These included *seeing integration as a continuous process that is never complete*; a *learning process of adopting new worldviews*; the *answering of questions the experience*

*brought up* or, in contrast, *successfully letting go of the experience*. For some participants their understanding of integration focused on alleviating the effects of their ontological shock, for example Beth describes creating a bridge between her metaphysical experiences and ordinary life:

> *Going from the mystical reality and the ordinary reality and yet bridging the gap, letting them co-exist. You know, I don't have to not have conversations about car insurance… like learning to deal with that side of things, like the yoga of going from the mystical to the mundane and learning to bridge that gap and the disparity between them and not feel like you're betraying either one side... I can live in both those sides at once now. I can go to work, and I can go and have a crazy trip and neither really imposes on the other.*

While many participants stressed that they are unsure if psychedelic integration is ever complete, ten participants explicitly noted they were content with their integration.

## Discussion

Previous studies have noted that psychedelics can lead to major shifts in identity and metaphysical beliefs [4,68]. This study is the first to explore *what it is like* to go through challenges that accompany such shifts, and *what helps* navigate persisting existential confusion. Our participants went through sometimes dramatic identity and belief shifts. Several described becoming obsessed with the meaning of their psychedelic experience, and eventually being helped by letting go of their need for answers. In all cases participants struggled with ontological insecurity; their psychedelic experiences overwhelmed their ability to make sense and as such trust the world around them, and this caused severe distress.

Participants described psychedelic experiences of ontological shock (supernatural, religious and spiritual experiences) coupled with core existential concerns (death, meaninglessness, responsibility, and aloneness). Stanislav Grof [53] has theorised that the ongoing distress that can follow psychedelic experiences arises from the unresolved confrontation with one's existential limits. The 'no-exit' situations identified by Grof also emerged as themes in our participants' accounts. The ontologically challenging nature of the experience was evidenced in participants' accounts of feeling fear, anxiety and confusion during the trip, and exiting it feeling overwhelmed, shaken and exhausted. Participants' experiences of ontological shock—marked by profound confusion, existential distress, and disorientation—can be understood as moments of 'liminality' [69], a state in which established beliefs and identities dissolve, opening individuals to new ways of seeing. These liminal states challenge individuals' sense of self and reality, creating a need for reorientation akin to the transformative learning process described by Buechner and colleagues [69]. Through this lens, we can see the profound implications of psychedelics as catalysts for transitions not only in beliefs but also in the ways individuals situate themselves within their reality, their existential identity.

The therapeutic benefits of psychedelics are theorised to be driven by increases in entropy –a measure of uncertainty– that shift individuals' reliance on their prior beliefs [36]. This destabilising mechanism allows for a recalibration of cognitive structures, enabling the reevaluation of previously rigid mental models. However, when individuals lack adequate psychological or social resources, this increase in uncertainty can lead to distress [70,71] manifesting as confusion and difficulty accommodating the ungrounding of established worldviews. The ensuing overwhelming sense of responsibility for meaning-making results in a sensation of 'groundlessness' [72]. This feeling of groundlessness, according to Meling [72], reflects the central underlying principle of cognition. Termed the 'foundationless foundation', it describes how humans, in a state of inherent uncertainty, enact and continuously reconstruct a world of meaning around them. Therefore, the dissolution of prior meaning frameworks under the influence of psychedelics can be seen as a direct encounter with this fundamental cognitive uncertainty. While such experiences underscore the sense of losing one's cognitive 'ground,' they also open avenues for exploring new existential territories. However, it is beyond the scope and aims of this study to delve into theological discussions of an ultimate ground, as proposed by thinkers like Tillich [56], which might provide a metaphysical

counterpoint to this existential uncertainty. Instead, we focus on psychological mechanisms with the aim of contributing to the identification of effective management of distress induced by ontologically challenging psychedelic experiences.

As Rodríguez Arce and Winkelman [73] argue, psychedelic experiences can trigger ontological shock that reflects in a loss of cognitive structuring and prior meaning frameworks. This risk may be especially pronounced in secularised contexts, where larger, communal meaning structures such as religion have largely dissolved [74]. In contrast, traditional psychedelic use within ritualised social contexts provides protective resources by offering a controlled environment and frameworks for integrating the experience [14,73]. The process of navigating ontological shock, as well as the loss and reconstruction of cognitive structures, may be heavily influenced not only by individuals' psychological resilience, but also by the availability of cultural resources.

According to Fromm [75], people subconsciously avoid the anxiety associated with the uncertainty of groundlessness by either adopting a rigid adaptation to life or transferring their freedom to ordering systems beyond their control. This need for order may relate to our participants' sometimes obsessive preoccupation with making sense of the experience and in turn to what the OCD International Foundation described as an 'Existential and Philosophical OCD' subtype [76]. Little research exists to-date about the distinct phenomenology and treatment of existential OCD [77], although Chawla and colleagues [78] had previously discussed evidence on the association between OCD symptom severity and the core existential concerns put forward by Yalom: death, freedom, isolation and meaninglessness, also seen in some of our participants' experiences. A few participants noted their prior 'questioning disposition' as a potential aetiological factor for their existential difficulties but the intersection of such factors and cultural frameworks are yet to be studied.

In line with evidence from Bremler and colleagues' [79] interview study, many participants described their experience as traumatic, with their lasting difficulties aligning significantly with DSM-5 criteria for PTSD. Grof [53] suggested that when individuals remain stuck in a liminal state after a challenging trip, the encountered material may resurface in various forms. Interviewees reported triggers that reignited their challenging psychedelic state, related to reminders of the trip's content or shifts into altered states of consciousness, such as falling asleep or meditating, when their vulnerability was heightened. Participants also experienced disruptions in daily life and work, including severe sleep disturbances—previously identified in studies as a common adverse effect of psychedelic use [80–82]. In these cases, using trauma-focused therapies, like EMDR, to process the psychedelic experience as a traumatic memory, may help resolve PTSD-like symptoms during the integration period [83].

Returning to the framing of psychedelic experience as a pivotal mental state [38], we highlight that these states are outcome-agnostic states — they can lead to disorientation or trauma if the developmental process is disrupted, or to psychological growth, through the process of integration. We propose that integrating ontologically challenging psychedelic experiences can be understood as a transformative learning process, wherein individuals are compelled to re-evaluate and reconstruct their frameworks for understanding reality and selfhood. Such experiences create profound disruptions, akin to Buechner and colleagues' [69] description of entering a liminal state—a phase marked by disorientation from prior knowledge and open to new ways of seeing the world. Participants in our study often reported facing 'existential struggle' as they navigated worldview shifts, questioning reality, identity, and purpose. These encounters align with transformative learning theorists' concept of "disorienting dilemmas"—events that destabilize previous certainties and open pathways for cognitive, emotional, and social growth. However, the pluripotentiality of these experiences means they are not inherently positive or therapeutic. As with any transformative learning process, the outcomes of psychedelic experiences depend heavily on the availability of supportive resources available to help individuals navigate and integrate these shifts.

Almost all participants mentioned navigating their difficulties through the use of practices and support that helped them 'ground': 'come back to [their] body' and let go of cognitive preoccupations about their experience. Notably, some participants described how cognitively distancing themselves from their experience helped them move on from troubled mind-states. Meditation was reported as helpful in cases where it aimed at focusing attention outside the self but as unhelpful when focused inwards leading participants to become further 'stuck in [their] head'. These findings, along with reports of

obsessive preoccupation suggest a cognitive overload that individuals attach significance to, but lack resources to navigate. The special role of 'grounding' for our participants can be understood through literature on trauma recovery where it is understood to help stabilise one's emotional state and bring focus back from dissociated states to one's body and the present moment [84]. Dissociation is an adaptive response that compartmentalises overwhelming experiences, storing memories in isolated fragments [85]. Grounding can reorient the person back to the present moment when dissociation arises and as such aid integration of the overwhelming experience.

The prevalent theme of dissociation manifesting as disembodiment, may be further understood as a response to being faced with one's existential vulnerability [86]. Binder [86] notes that bodily identity provides humans with a sense of continuity but also vulnerability to death. Being faced with one's mortality in a psychedelic experience can trigger feelings of ontological insecurity as an insecurity of being, a matter of life and death [60]. According to this lens, dissociation and feelings of being 'stuck outside of [one's] body' may serve as protective coping until the feelings of insecurity resolve and the individual is able to 'ground back'.

Some participants found relief in simply reframing their experience as an inner state rather than an ultimate truth about the universe. Others described engaging with structured ways of making sense. This was sometimes completely self-driven, such as through journaling and consciously breaking down the experience in small parts to process day by day. In many cases it was the adoption of established spiritual, religious or scientific frameworks that provided a new structure for making sense. Notably the framing of 'spiritual emergence' was found particularly helpful coupled with the notion that psychedelic-induced difficulties have significance and are part of a process of spiritual growth. A previous study on ayahuasca integration also highlighted the value of interpretive frameworks [43]. Explanatory systems, such as religious, scientific or spiritual framings provide normative explanations for unusual experiences. The normalisation of this otherwise confusing and isolating experience seemed to play an important role in participants' recovery whether through finding resonance in established frameworks, hearing about how others have suffered through similar challenges or speaking to friends and loved ones about their own challenges.

Interpersonal support was also often found through reaching out to community or finding a new community where psychedelic experience could be talked about openly. The elements of feeling seen, heard, believed and understood were mentioned as crucial to recovery. Connection with others and a like-minded community as a beneficial aid to integration was also a core finding in Cowley-Court and colleagues' study on ayahuasca integration [43] as well as our team's prior survey [44]. Some participants, however, were disturbed by elements of psychedelic-associated cultures which have a spirituality-focus but often promote spiritual bypassing [49,87] or separatist feelings of grandiosity based on a sense of spiritual superiority [88]. Interpersonal support which prevents the person from critically evaluating and coming to their own meaning can cause further harm.

We also asked participants for their lived-experience-informed learnings for the field. They noted the need for increasing awareness of harms but also of the potential to overcome challenging psychedelic experiences, in order to give hope to others who find themselves in their shoes. A need for structures that enable sharing and offer support during and after psychedelic experiences was identified, along with legalisation and decriminalisation policy changes to enable this. Participants' concerns reflected current wider debates in the field; the underestimation of harms and the commercialisation of psychedelics were discussed in relation to the indiscriminate hype that has characterised narratives around the re-emergence of psychedelic research [89].

In our study, participants highlighted grounding practices, social support, and cognitive reframing as essential resources for finding stability, making sense of their experiences, and re-establishing ontological security. Without these resources and support, individuals risk becoming mired in confusion, preoccupation, or existential despair. Viewing integration as a form of transformative learning underscores the importance of accessible frameworks and community-based support systems to facilitate this process. Such support can harness the potential of psychedelics as pivotal mental states (PiMS) that may foster psychological flexibility and personal growth, or, conversely, lead to prolonged distress if left unresolved.

Effective support structures can help individuals navigate and meaningfully integrate their experiences, ensuring these encounters contribute to psychological growth rather than becoming sources of lasting confusion.

Participants identified several factors as unhelpful in managing their existential struggles. While many found relief in spiritual frameworks, in other cases spiritual interpretations felt forced and instilled further fear. Attempts to escape through further substance use, including with cannabis, proved counterproductive for participants, as did an obsessive search for intellectual answers. Some participants found that revisiting the experience through additional psychedelic journeys allowed them to overcome the challenge, while others reported that these experiences were unhelpful or even harmful. Evans and Read [90] have previously suggested, based on a series of personal accounts, that individuals experiencing spiritual emergencies may benefit from revisiting altered states, but only after a significant period of time has passed and they feel more resilient and better able to cope.

Wide-ranging changes in metaphysical beliefs and associated challenges raise questions not only about the feasibility of obtaining informed consent prior to psychedelic experience [94] but also about the options one is left with to navigate their understanding. Towards the aim of better understanding these belief changes Rennie and colleagues' [91] created a Metaphysics Matrix Questionnaire (MMQ) as a measure of metaphysical beliefs. Sjöstedt-Hughes has further argued for the use of the MMQ as a 'menu' and the role of guides and facilitators in assisting the adoption of metaphysical interpretive frameworks following psychedelic experiences [92]. By contrast, Cheung and Yaden [93] caution towards supporting a meaning-making process driven by the individuals themselves. Notwithstanding, existential and spiritual elements appear to be crucial for psychedelic integration [45,94,95].

Many participants reported positive outcomes from navigating the challenges of their psychedelic experiences, including improved mental health, self-development and psychological growth. These findings align with concepts of post-traumatic growth [96] and narrative theories suggesting identity crises can trigger psychological transformation [97]. Following difficult or traumatic experiences, a period of self-questioning can catalyse the formation of a new sense of self, emerging with increased psychological maturity [98,99] that encourages prosociality [100]. Nearly half of the interviewees reflected on their increased compassion, appreciation of human connection, and motivation to support others linking these changes to their new understanding of themselves as part of the world; a recognition of shared human suffering, and alleviating their isolation through connection. Increased prosociality has also previously been linked to psychedelic use [101] and proposed as a key factor in facilitating wellbeing [102].

## Limitations and future directions

Our sample predominantly comprised participants from western cultures and only English speakers, who pointed to the lack of societal understanding of the spiritual nature of psychedelic experience. It may be that cultures with animist ontological frameworks have the structures to absorb ontological shock so that uncertainty is not experienced negatively. More research is needed to untangle the impact of such cultural framings and future studies should seek to broaden the cultural backgrounds explored.

Our study identified some helpful and harmful factors in navigating the implications of ontological shock. Considering ontological shock is often credited for beneficial outcomes, there is a need for more nuanced investigation of the complex interplay between belief shifts and mental health, including the role of uncertainty tolerance. Future research is needed to understand aetiological factors and the prevalence of these difficulties amongst psychedelic users as well as timescales for their development and helpful resources and support.

Many participants described their experiences as traumatic, reported symptoms of PTSD and dissociative experiences, describing a disconnection from their body. Indeed, several participants described being stuck in their heads, or outside their body. Future research could further explore dissociation spectrum challenges that arise from psychedelic experiences, how they may relate to traumatic encoding, existential vulnerability, or alternative framings of the experience, and aim to identify integration practices that restore a healthy mind-body connection.

The potential of psychedelic states to enable co-dependency and heighten interpersonal vulnerability was also brought forward by participants. Reports of abuse are increasingly coming out on media outlets [103]. Villiger and Trachsel [104] expanded on this in their ethical analysis of the malleability-inducing mechanisms of psychedelics that lead to the altering of beliefs. When people's meaning frameworks are 'ungrounded', they are left vulnerable to trusting authority, adopting new frameworks uncritically and ending up feeling more lost and confused. Further research is needed to understand what interpersonal and communal structures help people find meaning after psychedelic experiences and which types of support can be counterproductive.

## Conclusions

As psychedelic therapies and recreational use gain mainstream interest, establishing frameworks and resources to support ontological security is essential for mitigating distress and enhancing the transformative learning potential of psychedelic experiences. Our findings suggest that psychedelic-induced ontological shock leads to cognitive, emotional and social *ungrounding* which can in turn lead to prolonged existential confusion and challenges with meaning-making. *Grounding*, whether somatic, social, or through normalizing unusual experiences, was pivotal in helping individuals manage confusion, intense preoccupations, and feelings of isolation. Successful integration of ontologically challenging experiences appears to require re-establishing stability and meaning as individuals rebuild frameworks for understanding the world and their existential identity within it. Our study underscores the importance of grounding, meaning-making, and community support as foundational to the integration process, facilitating a pathway for transformative learning.

## Supporting information

Semi-structured Interview Guide. Table of all themes and subthemes with frequencies, percentages of sample and indicative quotes
(DOCX)

## Acknowledgments

EKA acknowledges the mutual learning and support nurtured through discussions with members of the Transdisciplinary Psychedelic Research Colloquium and the emerging community of practice within the Holistic Integration group in Exeter. We also thank Daniel Ingram for his support and his work through Emergence Benefactors and the EPRC.

## Author contributions

**Conceptualization:** Eirini K. Argyri, Jules Evans, David Luke, Katrina Michelle, Ed Prideaux, Ashleigh Murphy-Beiner, Oliver C. Robinson.

**Data curation:** Eirini K. Argyri, Jules Evans, Katrina Michelle, Cyrus Rohani-Shukla, Shayam Suseelan, Ed Prideaux, Oliver C. Robinson.

**Formal analysis:** Eirini K. Argyri, Jules Evans, Pascal Michael.

**Funding acquisition:** Jules Evans.

**Investigation:** Eirini K. Argyri, Katrina Michelle, Cyrus Rohani-Shukla, Shayam Suseelan, Ed Prideaux.

**Methodology:** Eirini K. Argyri, Oliver C. Robinson.

**Project administration:** Eirini K. Argyri.

**Supervision:** Oliver C. Robinson.

**Visualization:** Eirini K. Argyri, Rosalind McAlpine, Oliver C. Robinson.

**Writing – original draft:** Eirini K. Argyri, Jules Evans, David Luke, Pascal Michael, Oliver C. Robinson.

**Writing – review & editing:** Eirini K. Argyri, Jules Evans, David Luke, Pascal Michael, Katrina Michelle, Cyrus Rohani-Shukla, Shayam Suseelan, Ed Prideaux, Rosalind McAlpine, Ashleigh Murphy-Beiner, Oliver C. Robinson.

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
