## [Decision Letter · Decision Letter 0]

3 Sep 2024

PONE-D-24-20547Navigating Groundlessness: An interview study on dealing with ontological shock and existential distress following psychedelic experiencesPLOS ONE

Dear Dr. Ketzitzidou Argyri,

Thank you for submitting your manuscript to PLOS ONE. After careful consideration, we feel that it has merit but does not fully meet PLOS ONE’s publication criteria as it currently stands. Therefore, we invite you to submit a revised version of the manuscript that addresses the points raised during the review process.

The reviewers have completed their assessment of your paper. While they acknowledged its strengths, they also identified several areas that need correction. After reviewing the paper myself, I concur with the reviewers' evaluations and strongly encourage you to make the suggested revisions to improve the manuscript.

We look forward to receiving your revised manuscript.

Kind regards,

Michal Mahat-Shamir, Ph.D.

Academic Editor

PLOS ONE

Journal Requirements: When submitting your revision, we need you to address these additional requirements. 1. Please ensure that your manuscript meets PLOS ONE's style requirements, including those for file naming. The PLOS ONE style templates can be found at https://journals.plos.org/plosone/s/file?id=wjVg/PLOSOne_formatting_sample_main_body.pdf and https://journals.plos.org/plosone/s/file?id=ba62/PLOSOne_formatting_sample_title_authors_affiliations.pdf 2. We note that the grant information you provided in the ‘Funding Information’ and ‘Financial Disclosure’ sections do not match.  When you resubmit, please ensure that you provide the correct grant numbers for the awards you received for your study in the ‘Funding Information’ section. 3. Thank you for stating the following financial disclosure: "JE received funding from Emergence Benefactors and the William G. Nash Foundation.EKA is supported by the Economic and Social Research Council UK. "  Please state what role the funders took in the study.  If the funders had no role, please state: ""The funders had no role in study design, data collection and analysis, decision to publish, or preparation of the manuscript."" If this statement is not correct you must amend it as needed. Please include this amended Role of Funder statement in your cover letter; we will change the online submission form on your behalf. 4. We note that you have indicated that there are restrictions to data sharing for this study. For studies involving human research participant data or other sensitive data, we encourage authors to share de-identified or anonymized data. However, when data cannot be publicly shared for ethical reasons, we allow authors to make their data sets available upon request. For information on unacceptable data access restrictions, please see http://journals.plos.org/plosone/s/data-availability#loc-unacceptable-data-access-restrictions. Before we proceed with your manuscript, please address the following prompts: a) If there are ethical or legal restrictions on sharing a de-identified data set, please explain them in detail (e.g., data contain potentially identifying or sensitive patient information, data are owned by a third-party organization, etc.) and who has imposed them (e.g., a Research Ethics Committee or Institutional Review Board, etc.). Please also provide contact information for a data access committee, ethics committee, or other institutional body to which data requests may be sent. b) If there are no restrictions, please upload the minimal anonymized data set necessary to replicate your study findings to a stable, public repository and provide us with the relevant URLs, DOIs, or accession numbers. Please see http://www.bmj.com/content/340/bmj.c181.long for guidelines on how to de-identify and prepare clinical data for publication. For a list of recommended repositories, please see https://journals.plos.org/plosone/s/recommended-repositories. You also have the option of uploading the data as Supporting Information files, but we would recommend depositing data directly to a data repository if possible. Please update your Data Availability statement in the submission form accordingly. 5. Please include captions for your Supporting Information files at the end of your manuscript, and update any in-text citations to match accordingly. Please see our Supporting Information guidelines for more information: http://journals.plos.org/plosone/s/supporting-information.

Reviewers' comments:

Reviewer's Responses to Questions

**Comments to the Author**

1. Is the manuscript technically sound, and do the data support the conclusions?

Reviewer #1: Yes

Reviewer #2: Yes

2. Has the statistical analysis been performed appropriately and rigorously?

Reviewer #1: N/A

Reviewer #2: N/A

3. Have the authors made all data underlying the findings in their manuscript fully available?

Reviewer #1: Yes

Reviewer #2: Yes

4. Is the manuscript presented in an intelligible fashion and written in standard English?

Reviewer #1: Yes

Reviewer #2: Yes

5. Review Comments to the Author

Reviewer #1: This qualitative article deals with the experience of psychedelic use and outcomes in non therapeutic settings.

It was an interesting read, thought provoking and engaging in terms of experiences (non helpful experiences follwoign use was less interesting to me for some reason).

That being said, I felt that there was not a proficient lit. framework for discussing the types of effects of psychedelics on the human brain and especially the information processes that occur. Before discussing on page 2 “The process of making sense of and learning from psychedelic experiences, particularly those that are challenging, is commonly described as ‘integration’ [28,29], It would makes sense to write about information processing and how psychedelics affect this process (integration would be an outcome of information processing, of altering of consciousness and reworking of past concepts in ones mind following the experience).

The concept of ontological shock was put forward, but this shock is just one outcome altered information processing in the brain. I feel that processing models should be central in this discussion, especially since questions of being (ontological discourse- participants were recruited to share their experiences of "Existential Struggle,") were not the only experiences that were described by participants.

in other words, many of the experiences related to in the article by participants were not merely related to metaphysical beliefs but rather extreme experiences which were encoded as a threat to the individual, and this does not come out in the opening set up of the article.

What this article is inavertedly describing, in my eyes, is mainly overwhelming traumatic experiences/encoding within the psychedelic experience,

Note that some authors have offered (eg) that “It is possible that HPPD represents a form of traumatic anxiety disorder akin to PTSD [21, 25] or a form of health anxiety [26] triggered by the residual symptoms of the psychedelic experience.“

this logic can be put forward towards all the “residual” symptoms such as those described here are related to trauma related encoding, and not ontological issues.

https://doi.org/10.1371%2Fjournal.pone.0293349

BTW, In that study, 40% of the sample thought that a childhood trauma was implicated in experiencing post-psychedelic difficulties, which seems in line with what was described in this present article, regarding the pre-conditions of individuals participating in the study who may have been more likely to have adverse life experiences or a period of stress before the psychedelic experience, and thereby more prone to a traumatic reaction to the drug.

The “existential concerns” put forward here can easily be clustered under PTSD-like symptoms: and I therefore suggest trauma be taken more into consideration.

“Existential concern. For 9 individuals, this existential concern involved Perceived physical or ego death (death related - Criteria A in PTSD). The theme Overwhelming responsibility (thematically related to Guilt in EMDR therapy- Cognition and mood symptoms in PTSD) was conveyed by 7 of the participants. Experiences echoing a sense of Meaninglessness or emptiness (Again, Cognition and mood symptoms) were shared by 5 individuals. The final theme under this existential umbrella referred to Solipsistic isolation or Aloneness (again, - Cognition and mood symptoms),

To sum, trauma, information processing or consciousness models are lacking, in my eyes and provide a better framework, even if the ontological aspect is the main game.

Alternatively if existential struggle is indeed the sole the focus of this paper then more effort should be out into a conceptual model related to such a concept, and not just one paragraph on ontological shock (This is not my area of expertise, and more writing on the topic would be necessary).

See for example- some papers that are relevant .

Carhart-Harris R., Leech R., Hellyer P., Shanahan M., Feilding A., Tagliazucchi E., et al.. (2014b). The entropic brain: a theory of conscious states informed by neuroimaging research with psychedelic drugs. Front. Hum. Neurosci. 8:20. 10.3389/fnhum.2014.00020

Froese T. (2015). The ritualised mind alteration hypothesis of the origins and evolution of the symbolic human mind. Rock Art Res. 32, 90–97.

Gallimore A. (2015). Restructuring consciousness –the psychedelic state in light of integrated information theory. Front. Hum. Neurosci. 12:346 10.3389/fnhum.2015.00346

More in the - ITT

https://pubmed.ncbi.nlm.nih.gov/24811198/

see this for relevant in formation in ITT, effects of psychedelic, brain effects

https://www.frontiersin.org/journals/human-neuroscience/articles/10.3389/fnhum.2015.00346/full

(related to sexual abuse) Information processing of trauma. This can be an alternative framework, where the psychedelic experience can be regarded as an experience, which is not properly encoded in the brain, causing a posttraumatic reaction (as indicated in the paper)- then using this as a framework to describe how our brain means of make meaning of experiences.

EMDR theory discusses this in AIP terms, see F. Shapiro 2018 EMDR textbook, or an older trauma information model related to sexual abuse-

10.1016/0145-2134(93)90007-r

Page and Paragraph, additional comments

1 1 There are different types of uses of these substances, for instance- recreational, professional therapeutic, non professional therapeutic, what is meant by “use” ,- is there difference in the literature, because these participants were recreational self users (even if it was for self development intentions)

1 1 physicalism , non-physicalist beliefs – should give a context, definition, example.

1 2 Use a word other than trigger which may have a negative connotation.

2 2 Integration- define it in this context, find a conceptual model for integration (may be related to previous statement on information processing). for instance, Dan Seigel writes extensively around this issue, and it is interesting to understand how the different fields that he speaks about are related to psych. experiences.

7 Different nationalities, relate to this?

Life context- brain primed for turbulence, as stated above.

10 3 When describing experiences, add percentages not just numbers

10+ Was there a theme of bodily reactions/sensations? Described in exiting, does it have a space in itself even though these are commonly stated as a type of effect. The body is missing in this article.

22 last Self education is actually part of cognitive practices, mentioned before perhaps branch them together and differentiate between the two types (based on external information)

30 core existential concerns (death, meaninglessness, responsibility, and aloneness)= seem more like traumatic symptoms related to cluster 4- as I said above.

28 “ Self development and spiritual growth “ can be related to the concept of posttraumatic growth .

Using trauma-focused therapies, like EMDR to process the psychedelic experience as a traumatic memory, may help resolve PTSD-like symptoms during the integration period [71].

I agree.

Discussion was built better, but following a conceptual framework for the lit review, I would reorganize.

Reviewer #2: Overall Evaluation: The topic of the article is important and suitable for publication.

Introduction:

1) Please provide more detailed information on the types of psychedelic substances. Clarify the inclusion of cannabis (in the table for Cora & Don) and MDMA (for Ida), and explain why these are considered psychedelics.

2) Please expand on the ‘renaissance’ in psychedelic research.

3) While the research emphasizes the potential shortcomings and harms of psychedelic substances, the introduction should also briefly address the benefits that have contributed to the resurgence of interest in psychedelic research in recent years.

Method:

1) Include the reflexive aspect and researcher characteristics in the methodology, particularly in relation to their connection with psychedelic substances.

Table 1:

1) Age at Interview: For Aaron, the age range is listed as 45-55. If this is an estimation, please indicate that it is an assessment.

2) Inconsistencies in Age- Please correct these inconsistencies: Georgia is listed as 55 at the interview but 56 at the psychedelic experience. Similarly, Jessie is listed as 35 at the interview but 36 at the psychedelic experience. Chris is listed as 27 at the interview but 34 at the psychedelic experience.

3) For Youssef and Cora: The age ranges for their interviews and psychedelic experiences are unclear. Please clarify.

4) What substance is "Rape" (for Jessie)? Please provide clarification.

5) In the “Substance/s taken” column, does this indicate that the participants experienced difficulties after taking the listed psychedelic substances? For Aaron, you mention "different trips"—did he encounter difficulties during all of them? This needs clarification. And for Beth, is the reference to polysubstance use in one event? Similar clarification is needed for Cal and Clara.

Figures 1 and 2:

1) Figures 1 and 2 are somewhat blurry and difficult to read; please address this issue.

2) Titles should be added to the figures. Within the article text, include placeholders like “[Figure 1 insert here]” and “[Figure 2 insert here]” where appropriate.

6. PLOS authors have the option to publish the peer review history of their article (what does this mean? ). If published, this will include your full peer review and any attached files.

**Do you want your identity to be public for this peer review?** For information about this choice, including consent withdrawal, please see our Privacy Policy .

Reviewer #1: No

Reviewer #2: No

---

## [Author Response · Author response to Decision Letter 1]

10 Nov 2024

Response to Reviewers for Manuscript PONE-D-24-20547

Title: Learning from Groundlessness: An interview study on dealing with ontological shock and existential distress following psychedelic experiences

To: Dr. Michal Mahat-Shamir, Ph.D., Academic Editor

From: Eirini Ketzitzidou Argyri

Date: November 10th, 2024

Dear Dr. Mahat-Shamir,

Thank you for the opportunity to revise and resubmit our manuscript. We appreciate the detailed and thoughtful feedback from the reviewers, and we believe their comments have significantly helped improve the quality of our paper.

Below, we have provided a point-by-point response to each of the reviewers’ comments, outlining the corresponding changes made to the manuscript. The three required documents (a copy of this letter, the revised manuscript with tracked changes, the revised manuscript with accepted changes) have also been uploaded through the system. Please also note we have made a minor edit to the title to match the theoretical framework we added as part of the requested revisions.

First to address your points for additional information and clarification:

● PLOS ONE's style requirements: We have made amendments accordingly

● Funding: As requested we are copying the amended Funding statement here, including the role of the funders:

“JE received funding from Emergence Benefactors, the William G. Nash Foundation and the Sarlo Family. EKA also received funding from the Economic and Social Research Council UK. The funders had no role in study design, data collection and analysis, decision to publish, or preparation of the manuscript. "

● Data sharing restrictions:

The ethical approval from the University of Greenwich Ethics Committee (researchethics@greenwich.ac.uk) precludes sharing of the data. The full interview transcripts, although anonymised, contain potentially identifying information. Available data relevant to this study is presented in the manuscript text and in Supporting information.

● Captions for Supporting Information files: We have added those at the end of the manuscript

Responses to reviewers' comments:

Reviewer #1 points:

General

● Proficient literature framework discussing the effects of psychedelics on the human brain lacking, especially information processes

We thank the reviewer for the opportunity to revise and expand our theoretical framework. We make reference to existing theoretical framings for the action of psychedelic substances, specifically where they relate to belief changes, including the reviewers recommendations. However we also note we are not neuroscientists and this is a phenomenological paper, as such brain processing theories are beyond the focus of the paper and we don’t expand on them. We did take the opportunity to dive deeper into the literature that is fundamental to the existential focus of the paper and we are grateful to the reviewer for their suggestions.

Specifically we have strengthened, throughout the manuscript, both our discussion of existential literature, the potential psychological mechanisms where they relate to beliefs and help explain existential distress, framings of trauma (see below), and crucially our proposition of the concept of integration as transformative learning that follows pivotal mental states and the encounter with core existential concerns (ontological insecurity).

● More consideration on framing of trauma

Thank you for the pointers. Indeed many of our participants described their experiences as traumatic and their difficulties could be conceptualised within the clusters of ptsd criteria. We had originally considered this framework and moved away due to agreeing that what binds the findings instead is an existential focus and challenges with meaning making that relate to the ungrounding of prior beliefs and facing one's existential vulnerability. However, we have now expanded our discussion of trauma literature where relevant in both the introduction and discussion, acknowledging that trauma is amongst the potential outcomes and such framing, where it resonates with individuals, may also have therapeutic value in the management of associated distress.

● More literature on existential framing if this is the focus

See above, we have made considerable revisions to expand on this throughout the paper. Thank you for the invitation to do so.

Intro

● 1 1 What is meant by “use”

Clarification added.

● 1 1 physicalism , non-physicalist beliefs – should give a context, definition, example.

Definitions added.

● 1 2 Use a word other than trigger which may have a negative connotation.

Replaced with catalyse.

● 2 2 Integration- define it in this context, consider Dan Siegel’s work

We have now expanded our discussion on the concept of integration and make note of Dan Siegels work beyond psychedelics, thank you for the pointer. We feel our revisions and expanded discussion on the concept of integration has improved the paper.

● 7 “Different nationalities, relate to this?”

We are not fully clear on what the reviewer is requesting. However given our sample size we are not in a position to compare our findings based on nationality. We do make the suggestion in the discussion for future studies to explore the role of cultural background. The table is provided for descriptive demographic info of our participants, not for analytical focus.

● 10 3 When describing experiences, add percentages not just numbers

Thank you for this feedback. In qualitative analysis, we aim to explore the depth and complexity of participants’ experiences rather than quantifying responses. While participant counts for each theme are provided to offer transparency, percentages could imply representativeness that may not align with qualitative goals (see Braun & Clarke, 2006). Furthermore, we have included percentages in the supplementary materials themes table, where they can be referenced as needed without impacting the interpretative focus of the main text.

● 10+ Was there a theme of bodily reactions/sensations? Described in exiting, does it have a space in itself even though these are commonly stated as a type of effect. The body is missing in this article.

The body is missing in the article as the focus (including of the interview questions) was on meaning making. Indeed many participants struggled with feeling disconnected from their body, stuck in their heads, hence the need for grounding. We have added discussion of this in the Discussion, including relevance to dissociation, trauma and relevant existential interpretation.

● 22 last Self education is actually part of cognitive practices, mentioned before perhaps branch them together and differentiate between the two types (based on external information)

We appreciate you noticing the ambiguity on this part of the results, we have resolved this.

● 30 core existential concerns (death, meaninglessness, responsibility, and aloneness)= seem more like traumatic symptoms related to cluster 4- as I said above.

As above, we take your point and have returned to this in the discussion, thank you.

● 28 “ Self development and spiritual growth “ can be related to the concept of posttraumatic growth .

Yes, thank you. We make further reference to interpretations of posttraumatic growth and link to the framing of integration as transformative learning that we propose.

● Using trauma-focused therapies, like EMDR to process the psychedelic experience as a traumatic memory, may help resolve PTSD-like symptoms during the integration period [71]. I agree. Discussion was built better, but following a conceptual framework for the lit review, I would reorganize.

Thank you for noting this, we have reorganised and expanded the Discussion, as above.

Reviewer #2:

Introduction

● 1) Please provide more detailed information on the types of psychedelic substances. Clarify the inclusion of cannabis (in the table for Cora & Don) and MDMA (for Ida), and explain why these are considered psychedelics.

Thank you for pointing this, clarification added.

● 2) Please expand on the ‘renaissance’ in psychedelic research.

We take your point, we have rephrased to resurgence of research and defined it in time.

● 3) While the research emphasizes the potential shortcomings and harms of psychedelic substances, the introduction should also briefly address the benefits that have contributed to the resurgence of interest in psychedelic research in recent years.

This has now been addressed in edits to the opening paragraphs.

Method:

● 1) Include the reflexive aspect and researcher characteristics in the methodology, particularly in relation to their connection with psychedelic substances.

Thank you for the suggestion, we have now added a section for reflexivity in the Method to clarify our positioning and process.

Table 1:

● 1) Age at Interview: For Aaron, the age range is listed as 45-55. If this is an estimation, please indicate that it is an assessment.

We have added additional clarification.

● 2) Inconsistencies in Age- Please correct these inconsistencies: Georgia is listed as 55 at the interview but 56 at the psychedelic experience. Similarly, Jessie is listed as 35 at the interview but 36 at the psychedelic experience. Chris is listed as 27 at the interview but 34 at the psychedelic experience.

Thank you for spotting these inconsistencies, they have been corrected.

● 3) For Youssef and Cora: The age ranges for their interviews and psychedelic experiences are unclear. Please clarify.

Clarified as above.

● 4) What substance is "Rape" (for Jessie)? Please provide clarification.

Clarified as above.

● 5) In the “Substance/s taken” column, does this indicate that the participants experienced difficulties after taking the listed psychedelic substances? For Aaron, you mention "different trips"—did he encounter difficulties during all of them? This needs clarification. And for Beth, is the reference to polysubstance use in one event? Similar clarification is needed for Cal and Clara.

Thank you for catching this ambiguity. Clarified as above.

Figures 1 and 2:

● 1) Figures 1 and 2 are somewhat blurry and difficult to read; please address this issue.

Addressed.

● 2) Titles should be added to the figures. Within the article text, include placeholders like “[Figure 1 insert here]” and “[Figure 2 insert here]” where appropriate.

We have added these.

Thank you for considering this revised version of our work. We are confident that the improvements we have made in response to the reviewers' insightful suggestions contribute to a more robust and impactful manuscript. We look forward to your feedback and your decision regarding publication of our study in PLOS One.

Sincerely,

Eirini K. Argyri

University of Exeter

---

## [Decision Letter · Decision Letter 1]

23 Mar 2025

Learning from Groundlessness: An interview study on dealing with ontological shock and existential distress following psychedelic experiences

PONE-D-24-20547R1

Dear Dr. Ketzitzidou Argyri,

We’re pleased to inform you that your manuscript has been judged scientifically suitable for publication and will be formally accepted for publication once it meets all outstanding technical requirements.

Kind regards,

Michal Mahat-Shamir, Ph.D.

Academic Editor

PLOS ONE

Additional Editor Comments (optional):

Reviewers' comments:

Reviewer's Responses to Questions

**Comments to the Author**

1. If the authors have adequately addressed your comments raised in a previous round of review and you feel that this manuscript is now acceptable for publication, you may indicate that here to bypass the “Comments to the Author” section, enter your conflict of interest statement in the “Confidential to Editor” section, and submit your "Accept" recommendation.

Reviewer #1: All comments have been addressed

Reviewer #2: All comments have been addressed

2. Is the manuscript technically sound, and do the data support the conclusions?

Reviewer #1: Yes

Reviewer #2: Yes

3. Has the statistical analysis been performed appropriately and rigorously?

Reviewer #1: I Don't Know

Reviewer #2: Yes

4. Have the authors made all data underlying the findings in their manuscript fully available?

Reviewer #1: Yes

Reviewer #2: Yes

5. Is the manuscript presented in an intelligible fashion and written in standard English?

Reviewer #1: Yes

Reviewer #2: Yes

6. Review Comments to the Author

Reviewer #1: Most suggested changes were made.

One additional change that shpuld be made:

Just as traumatic experiences can lead to posttraumatic stress disorder (PTSD), post-traumatic growth (PTG), (Add: or both + a citation.)

Reviewer #2: After reviewing the revised manuscript and considering the authors' responses to my previous comments, I find that all concerns and suggestions have been adequately addressed to my satisfaction. The authors have made the necessary revisions, improving the clarity, and adding reflexive aspects and researcher characteristics in the methodology.

Given these improvements, I recommend accepting the manuscript for publication.

7. PLOS authors have the option to publish the peer review history of their article (what does this mean? ). If published, this will include your full peer review and any attached files.

**Do you want your identity to be public for this peer review?** For information about this choice, including consent withdrawal, please see our Privacy Policy .

Reviewer #1: No

Reviewer #2: No

---

## [Editor Report · Acceptance letter]

PONE-D-24-20547R1

PLOS ONE

Dear Dr. Argyri,

I'm pleased to inform you that your manuscript has been deemed suitable for publication in PLOS ONE. Congratulations! Your manuscript is now being handed over to our production team.

Kind regards,

on behalf of

Prof. Michal Mahat-Shamir

Academic Editor

PLOS ONE